# Provable Hierarchical Lifelong Learning with a Sketch-based Modular Architecture

## Abstract

We propose a modular architecture for lifelong learning of hierarchically structured tasks. Specifically, we prove that our architecture is theoretically able to learn tasks that can be solved by functions that are learnable given access to functions for other, previously learned tasks as subroutines. We empirically show that some tasks that we can learn in this way are not learned by standard training methods in practice; indeed, prior work suggests that some such tasks cannot be learned by *any* efficient method without the aid of the simpler tasks. We also consider methods for identifying the tasks automatically, without relying on explicitly given indicators.

## 1 Introduction

How can complex concepts be learned? Human experience suggests that hierarchical structure is key: the complex concepts we use are no more than simple combinations of slightly less complex concepts that we have already learned, and so on. This intuition suggests that the learning of complex concepts is most tractably approached in a setting where multiple tasks are present, where it is possible to leverage what was learned from one task in another. Lifelong learning (Silver et al., 2013; Chen & Liu, 2018) captures such a setting: we are presented with a sequence of learning tasks and wish to understand how to (selectively) transfer what was learned on previous tasks to novel tasks. We seek a method that we can analyze and prove leverages what it learns on simple tasks to efficiently learn complex tasks; in particular, tasks that could not be learned without the help provided by learning the simple tasks first.

In this work, we propose an architecture for addressing such problems based on creating new modules to represent the various tasks. Indeed, other modular approaches to lifelong learning (Yoon et al., 2018; Rusu et al., 2016) have been proposed previously. But, these works did not consider what we view as the main advantage of such architectures: their suitability for theoretical analysis. We prove that our architecture is capable of efficiently learning complex tasks by utilizing the functions learned to solve previous tasks as components in an algorithm for the more complex task. In addition to our analysis proving that the complex tasks may be learned, we also demonstrate that such an approach can learn functions that standard training methods fail to learn in practice, including some that are believed not to be learnable, even in principle (Klivans & Sherstov, 2009). We also consider methods for automatically identifying whether a learning task posed to the agent matches a previously learned task or is a novel task.

We note briefly that a few other works considered lifelong learning from a theoretical perspective. An early approach by Solomonoff (1989) did not seriously consider computational complexity aspects. Ruvolo & Eaton (2013) gave the first provable lifelong learning algorithm with such an analysis. But, the transfer of knowledge across tasks in their framework was limited to feature learning. In particular, they did not consider the kind of deep hierarchies of tasks that we seek to learn.

### 1.1 Overview of the architecture

The main technical novelty in our architecture over previous modular lifelong learners is that ours uses a particular type of internal data structure called *sketches* (Ghazi et al., 2019; Panigrahy, 2019). All such data, including inputs from the environment, outputs from a module for another task, decisions such as choosing an action to take, or even descriptions of the modules themselves, are

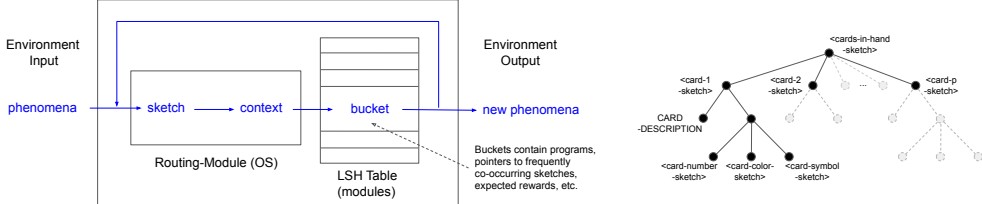

Figure 1: *(left)* The Routing-module (OS) routes sketches to the programs in the LSH table, which in turn produces sketches that are fed back to the OS in addition to sketches of inputs from the environment. The OS, while shown here as a distinct module, could itself be a module (program) in the LSH hash table. *(right)* Sketch of a hand of cards during a card game. The ⟨cards-in-hand-sketch ⟩is a tuple of $p$ sub-sketches (one for each card), and each ⟨card-$i$-sketch ⟩is itself a compound sketch: for example, ⟨card-1-sketch ⟩consists of the CARD-DESCRIPTION label/type and three sub-sketches describing the card's number, color, and symbol.

encoded as such sketches. Although sketches have a dense (vector) representation, they can also be interpreted as a kind of structured representation (Ghazi et al., 2019, Theorem 9) and are *recursive*; that is, they point to the previous modules/events that they arose from (Figure 1, right). However, in order to construct these sketches in Ghazi et al. (2019), the structure of the network is assumed to be given. No algorithms for constructing such a hierarchical network of modules from training data were known. In this work we show a method to construct such a hierarchical network from training data. We provide an architecture and algorithms for learning from a stream of training inputs that produces such a network of modules over time. This includes challenges of identifying each module, and discovering which other modules it depends on.

Our architecture can be viewed as a variant of the Transformer architecture (Radford et al., 2021; Shazeer et al., 2017), particularly the Switch Transformer (Fedus et al., 2021) in conjunction with the idea of Neural Memory (Wang et al., 2021). Instead of having a single feedforward layer, the Switch Transformer has an array of feedforward layers that an input can be routed to at each layer. Neural Memory on the other hand is a large table of values, and one or a few locations of the memory can be accessed at each layer of a deep network. In a sense the Switch Transfomer can be viewed as having a memory of possible feedforward layers (although they use very few) to read from. It is viewing the memory as holding "parts of a deep network" as opposed to data, although this difference between program and data is artificial: for example, embedding table entries can be viewed as "data" but are also used to alter the computation of the rest of the network, and in this sense act as a "program modifier".

The key component of our architecture is a locality sensitive hash (LSH) table based memory (see Wang et al. (2021)) that holds sketches of data (such as inputs) and *modules* or programs (think of an encoding of a small deep network) that handles such sketches (Figure 1, left). The "operating system" of our architecture executes the basic loop of taking sketches (either from the environment or from internal modules) and routing/hashing them to the LSH table to execute the next module that processes these sketches. These modules produce new sketches that are fed back into the loop.

New modules (or concepts) are formed simply by instantiating a new hash bucket whenever a new frequently-occurring context arises, i.e. whenever several sketches hash to the same place; the context can be viewed as a function-pointer and the sketch can be viewed as arguments for a call to that function. Frequent subsets of sketches may be combined to produce *compound* sketches. Finally we include pointers among sketches based on co-occurrence and co-reference in the sketches themselves. These pointers form a knowledge graph: for example if the inputs are images of pairs of people where the pairs are drawn from some latent social network, then assuming sufficient sampling of the network, this network will arise as a subgraph of the graph given by these pointers. The main loop allows these pointers to be dereferenced by passing them through the memory table, so they indeed serve the intended purpose.

The main idea of the architecture is that all events produce sketches, which can intuitively be thought of as the "mind-state" of the system when that event occurs. The sketch-to-sketch similarity property (see below) combined with a similarity-preserving hash function ensures that similar sketches go

to the same hash bucket (Appendix B); thus the hash table can be viewed as a content addressed memory. See Figure 1 for an illustration of this. We remark that the distances between embeddings of scene representations were used to automatically segment video into discrete events by Franklin et al. (2020), and obtained strong agreement with human annotators. The thresholded distance used to obtain the segmentation is analogous to our locality-sensitive hashes, which we use as context sketches.

A sketch can be viewed at different levels of granularity before using it to access the hash table; this becomes the *context* of the sketch. Each bucket contains a program that is executed when a sketch arises that indexes into that bucket. The program in turn produces outputs and new sketches that are routed back to the hash table. The system works in a continuous loop where sketches are coming in from the environment and also from previous iterations; the main structure of the loop is:

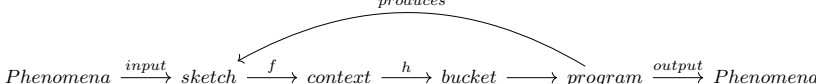

$$Phenomena \xrightarrow{input} sketch \xrightarrow{f} context \xrightarrow{h} bucket \longrightarrow program \xrightarrow{output} Phenomena$$

Thus external and internal inputs arrive as sketches that are converted into a coarser representation using a function $f$ (see Section 2.1 below) and then hashed to a bucket using a locality-sensitive hash function $h$. The program at that bucket is executed to produce an output-sketch that is fed back into the system and may also produce external outputs. This basic loop (described in Algorithm 1) is executed by the routing module, which can be thought of as the operating system of the architecture.

## 2 SKETCHES REVIEW

Our architecture relies on the properties of the sketches introduced in Ghazi et al. (2019). In this section we briefly describe the key properties of these sketches; the interested reader is referred to Ghazi et al. (2019); Wang et al. (2021) for details.

A *sketch* is a compact representation of a possibly exponentially-wide ($d \times N$) matrix in which only a small number $k$ of the columns are nonzero, that supports efficient computation of inner products, and for which encoding and decoding are performed by linear transformations. For concreteness, we note that sketches may be computed by random projections to $\mathbb{R}^{d'}$ for $d' \geq kd \log N$; the Johnson-Lindenstrauss Lemma then guarantees that inner-products are preserved.

For our purposes, we suppose modules $M_1, \ldots, M_N$ produce vectors $x_1, \ldots, x_N \in \mathbb{R}^d$ as output, where only $k$ of the modules produce (nonzero) outputs. We view the sparse collection of module outputs as a small set of pairs of the form $\{[M_{i_1}, x_{i_1}], \ldots, [M_{i_k}, x_{i_k}]\}$: For example an input image has a sketch that can be thought of as a tuple [IMAGE, ⟨bit-map-sketch⟩]. An output by an image recognition module that finds a person in the image can be represented as [PERSON, [⟨person-sketch⟩, ⟨position-in-image-sketch⟩]); here IMAGE and PERSON can be thought of as "labels". If the outputs of these modules are vector embeddings in the usual sense, then indeed the inner products measure the similarity of the objects represented by the output embeddings.

Observe that the constituent individual vectors $x_j$ in a sketch may themselves be sketches. For example, ⟨person-sketch⟩ could in turn be set of such pairs {[NAME,⟨name-sketch⟩], [FACIAL-FEATURES,⟨facial-sketch⟩], [POSTURE,⟨posture-sketch⟩]}, and an image consisting of multiple people could be mapped by our recognition module to a set {⟨person-1-sketch⟩, ⟨person-2-sketch⟩,...,⟨person-$k$-sketch⟩}. Note if the if the tuple is very large, we will not be able to recover the sketch of each of its members but only get a "average" or summary of all the sketches – however if a member has high enough relative weight (see (Ghazi et al., 2019, Section 3.3)) it can be recovered. Appendix C.1 discusses how large objects can be stored as sketches hierarchically.

Indeed, following Ghazi et al. (2019), the outputs of modules in our architecture will be tuples that, in addition to an "output" component, represent the input sketches which, in turn, represent the modules that produced those inputs, e.g., {[MODULE-ID,⟨module-id⟩], [OUTPUT-SKETCH,⟨output-sketch⟩], [RECURSIVE-INPUT-SKETCH, ⟨recursive-input-sketch⟩]}. By recursively unpacking the input sketch, it is possible to reconstruct the history of computation that produced the sketch.

---

**Algorithm 1** Informal presentation of the main execution loop

---

**Input:** input sketch $T$ (this sketch may contain a desired output for training)

1  current-sketches $\leftarrow \{T\}$
2  **while** current-sketches *is not empty*:
3    current-programs $\leftarrow \emptyset$
4    **foreach** *sketch $S$ in* current-sketches **do**
5     extract context $C = f(S)$
6     update access-frequency-count of bucket $h(C)$
7     **if** *bucket $h(C)$ has a program $P$*:
8      append $(S, P)$ to current-programs
9     **else:**
10     **if** *bucket $h(C)$ is frequently accessed*:
11      initialize program at $h(C)$ with some random program and mark it for training.
12      Fetch programs $P_i$ (possibly by some similarity criterion), append those $(S, P_i)$ to current-programs
13   Routing module chooses some subset of current-programs, runs each program on its associated sketch, appends outputs to current-sketches
14   Append sketches on outgoing edges of accessed buckets to current-sketches
15   **if** *any of the programs are marked for training*:
16    routing module picks one or some of them and trains them, and may choose to stop execution loop
17   **if** *any of the sketches is of (a special) type OUTPUT sketch*:
18    routing module picks one such, outputs that sketch or performs that action, and may choose to stop execution loop
19   **if** *any of the sketches is of type REWARD sketch (say for correct prediction or action)*:
20    updates the reward for this bucket and propagates those rewards to prior buckets
21   Routing module picks $k$ combinations of sketches in current-sketches, and combine them into compound sketches: $S_1, \ldots, S_k$ (may produce 0 sketches)
22   current-sketches $\leftarrow \{S_1, \ldots, S_k\}$

---

## 2.1 PRINCIPLES OF THE ARCHITECTURE

The following are the guiding principles behind the architecture.

1.  **Sketches.** All phenomena (inputs, outputs, commonly co-occurring events, etc) are represented as sketches. There is a *function from sketch to context $f : S \to C$* that gives a coarse grained version of the sketch. This is obtained by looking at the fields in the sketch $S$ that are high level labels and dropping fine details with high variance such as attribute values; it essentially extracts the "high-level bits" in the sketch $S$.

2.  **Hash table indexed by context that is robust to noise.** *(more details in Appendix C.2)* The hash function $h : C \to$ [hash-bucket] is "locality sensitive" in the sense that similar contexts are hashed to the same bucket with high probability. Each hash bucket may contain a trainable program $P$, and summary statistics as described in Figure 3. We don't start to train $P$ until the hash bucket has been visited a sufficient number of times. (Note: A program may not have to be an explicit interpretable program but could just be an "embedding" that represents (or modifies) a neural network.)

3.  **Routing-module.** *(can be implemented by Alg.2 )* Given a set of sketches from the previous iteration, the routing module identifies the top ones, applies the context function $f$ followed by the hash function $h$ to route them to their corresponding buckets. Before feeding these to $f$, the routing module picks a subset of sketches and combines them into compound sketches. The routing module makes all subset-choosing decisions

In addition, we can also keep associations of frequently co-occuring sketch contexts as edges across buckets forming knowledge graph. Please see Appendix C and G for details.

Thus new modules (or concepts) are formed simply by a new frequently occurring context (see earlier papers on how sketches are stored in LSH based sketch memory). Since sketches are fed to the programs indexed by their context, the context can be viewed as a function-pointer and the

sketch can be viewed as arguments for a call to that function; multiple arguments can be passed by using a compound sketch. Programs that call other modules can be represented as a computation DAG over modules at the nodes .

## 3    INDEPENDENT TASKS AND ARCHITECTURE V0

Our learning problem follows a formulation similar to Ruvolo & Eaton (2013). In a lifelong learning system, we are facing a sequence of supervised learning tasks: $\mathcal{Z}^{(1)}, \ldots, \mathcal{Z}^{(T_{max})}$. In contrast to Ruvolo & Eaton, at each step we will generally obtain a single input (in the form of sketch $(s_t, \mathbf{x}^{(t)}, \mathbf{y}^{(t)})$) that contains DATA $\mathbf{x}^{(t)} \in \mathcal{X}^{(t)}$, TARGET $\mathbf{y}^{(t)} \in \mathcal{Y}^{(t)}$ and task descriptor sketch (vector) $s_t$, where the $t$th task is given by a hidden function $\hat{\phi}^{(t)} : \mathcal{X}^{(t)} \to \mathcal{Y}^{(t)}$ that we want to learn, and $\hat{\phi}^{(t)}(\mathbf{x}^{(t)}) = y^{(t)}$. We assume that that the tasks are uniformly distributed, and the distributions over task data are stationary: i.e., at each step, the task is sampled uniformly at random, and for the sampled task $t$, the data is sampled independently from a fixed distribution on $\mathcal{X}^{(t)}$ for $t$. In this setting, we assume the task functions are all members of a common, known class of functions $\mathcal{M}$ for which there exists an efficient learning algorithm $\mathcal{A}_{\mathcal{M}}$, i.e., $\mathcal{A}_{\mathcal{M}}$ satisfies a standard PAC-learning guarantee: when provided with a sufficiently large number of training examples $M$, with probability $1 - \delta$ $\mathcal{A}_{\mathcal{M}}$ returns a function that agrees with the task function with probability at least $1 - \epsilon$ on the task distribution. For example, SGD learns a certain class of neural networks $\mathcal{M}$ with a small constant depth. Indeed, we stress that this setting does not require transfer across tasks.

Our architectures are instantiated by a choice of hash function $h$ and a context function $f$. Architecture v0 uses a very simple context function $f$: it projects the sketch down to the task descriptor $t$, dropping the DATA $\mathbf{x}^{(t)}$ and TARGET $y^{(t)}$ components. (Other combinatorial decisions in the routing module are NO-OPs.) Each time we receive an input learning sample, we will call Alg. 1 (input is a single sketch).

**Claim 3.1.** *Given an error rate $\epsilon > 0$ and confidence parameter $\delta > 0$ and $N$ independent tasks, each of which require at most $M = M(\epsilon, \delta)$ examples to learn to accuracy $1 - \epsilon$ with probability $1 - \delta$, and training data as described in above, with probability $1 - (N+1)\delta$, Architecture v0 learns to perform all $N$ tasks to accuracy at least $1 - \epsilon$ in $O(MN \log \frac{N}{\delta})$ steps.*

## 4    HIERARCHICAL LIFELONG LEARNING AND ARCHITECTURE V1

We follow a similar problem formulation as in Sec.3, but in this case a task can depend on other tasks. We assume that the structure of dependencies can be described by a degree-$d$ directed acyclic graph (DAG), in which the nodes correspond to tasks. Each task $t$ depends on at most $d$ other tasks $t'_1, \ldots, t'_d$, indicated by the nodes in the DAG with edges to its node, and the task is to compute the corresponding function $\hat{\phi}^{(t)} = \phi^{(t)}(\hat{\phi}^{(t'_1)}, \ldots, \hat{\phi}^{(t'_d)})$ where $\phi^{(t)} \in \mathcal{M}$. If $t'_1, \ldots, t'_d$ are sources in the DAG (no incoming edges) then $\hat{\phi}^{(t'_i)} \in \mathcal{M}$. We assume that all tasks share a common input distribution. We will call the functions from $\mathcal{M}$ *atomic modules*, since they are the building blocks of this hierarchy. We will call functions that call other functions in the DAG, such as $\hat{\phi}^{(t)}$ above, a *compound module*. As before, we assume $\mathcal{M}$ is a learnable function class. However, $\hat{\phi}^{(t)}$ might not belong to a learnable function class due to its higher complexity. Here, we will assume moreover that the algorithm $\mathcal{A}_{\mathcal{M}}$ for learning $\mathcal{M}$ is robust to label noise. Concretely, we will assume that if an $\epsilon$-fraction of the labels are corrupted by an adversary, then $\mathcal{A}_{\mathcal{M}}$ produces an $O(\epsilon)$-accurate function. We note that methods are known to provide SGD with such robustness for strongly convex loss functions, even if the features are corrupted during training (Diakonikolas et al., 2019) (see also, e.g., Li et al. (2020); Shah et al. (2020)). In this setting, we assume that the tasks are again sampled uniformly at random, and that the data is sampled independently from a common, fixed distribution for all tasks.

As with the architecture v0, v1 uses any locality sensitive hash function $h$ and a context function $f$ that projects the input sketch down to the task descriptor, discarding other components. The primary modifications are that

1. v1 tracks whether tasks are "well-trained," freezing their parameters when they reach a given accuracy for their level of the hierarchy, and

2. until a "well-trained" model is found, we train candidate models for the task in parallel that use the outputs of each possible subset of up to $d$ well-trained modules as auxiliary inputs.

We will maintain a global task level $L$, initially $0$. We define the target accuracy for level-$L$ tasks to be $\epsilon_L = (2dC)^L \epsilon$, where $C$ is the constant under the big-O for the guarantee provided by our robust training method; we let $M_L$ denote the sample complexity of robustly learning members of our class $\mathcal{M}$ to accuracy $1 - Cd\epsilon_{L-1}$ with confidence $1 - \delta$ when a $1 - d\epsilon_{L-1}$-accurate model exists. We check if any tasks became well-trained in level $L - 1$, and if so, for all tasks that are not yet well-trained, we initialize models for all combinations of up to $d - 1$ other well-trained tasks for each such new task. Each model is of the form $\phi(\hat{\phi}_{i_1}(x), ..., \hat{\phi}_{i_k}(x))$, where $i_1, \ldots i_k$ ($k \leq d$) is the corresponding subset of well-trained tasks such that at least one has level $L - 1$. On each iteration, the arriving example is hashed to a bucket for task $t$. We track the number of examples that have arrived for $t$ thus far at this level. For the first $M'$ examples that arrive in a bucket, we pass the example to the training algorithms for each model for this task, which for example completes another step of SGD. Once $M_L$ examples have arrived, we count the fraction of the next $O(\frac{d}{\epsilon_L} \log \frac{N}{\delta})$ examples that are classified correctly by each of the models. We thus check if its empirical accuracy is guaranteed to be at least $1 - \epsilon_L$ with high probability. If the empirical accuracy is sufficiently high, we mark the task as well-trained and use this model for the task, discarding the rest of the candidates. Once all of the tasks are well-trained or have obtained $M_L + O(\frac{d}{\epsilon_L} \log \frac{N}{\delta})$ examples since the global level incremented to $L$, we increment the global level to $L + 1$.

**Lemma 4.1.** *Suppose at each step, a task $t$ is chosen uniformly random from the set of tasks $\{t_1, \ldots, t_N\}$ in a DAG of height $\ell$, along with one random sample $(x, y)$ where $\hat{\phi}^{(t)}(x) = y$. Then after $\ell M N \ln(1/\delta)$ steps all the tasks will be well-trained (training error rate $\leq \epsilon_L$ for each module at level $L$) w.h.p. We will call SGD $O(\ell M N^{(1+d)} \ln(1/\delta))$ times during the training. Here, $M$ is the upper bound of all $M_L$.*

In the above discussion we argued at an algorithmic level and ignored the specific architecture details of which buckets the $\binom{N}{d}$ candidate modules are trained and how eventually a single compound module gets programmed in the bucket $h(s_t)$. See Appendix E for those details.

# 5 ARCHITECTURE V2: TASKS WITHOUT PRECISE EXPLICIT DESCRIPTIONS

We follow a similar problem formulation as in Sec. 4. However now clear task descriptors may not be provided externally, but may implicitly depend on the output of a previous module. (detailed examples in Appendix.F ). The following definitions and assumption differ from Sec. 4.

**Definition 5.1** (Tasks). Let $\mathcal{U}$ be a space of all (potentially recursive) sketches that include the input and output of all modules. ($\mathcal{U}$ can be polymorphic, that is, it can contain multiple different data types). Each task $t_i$ is a mapping $: \mathcal{U} \to \mathcal{U}$. The input distribution of $t_i$, $\mathcal{D}_i$, is supported on $\mathcal{U}$.

**Definition 5.2** (Latent dependency DAG). The latent *dependency* DAG is a DAG with nodes corresponding to tasks $t_1, .., t_N$ and edges indicating dependencies. Each task at an internal node depends on at most $d$ other tasks ($d$ may not be known to the learner, but is a small quantity).

**Definition 5.3** (Latent circuit). Given a dependency DAG, for each task $t_i$ there is a latent circuit with gates (nodes) corresponding to the tasks $t_i'$ that it depends on. In this circuit for $t_i$, there are (potentially) multiple sinks (nodes with no outgoing edges). The output of these sinks will be the inputs to some atomic module, which gives the output of $t_i$. There are multiple atomic internal modules for each $t_i$ and the circuit routes each example to one of these modules. Each $t_i$ is "vague" in the sense that there are multiple modules that can cater to an example of this task.

**Definition 5.4** (Hidden task description / Context). Given the circuit of each $t_i$, there is a fixed (unknown) subset of the outputs of the circuit that give a context value that uniquely identifies $t_i$. There exists a bound $g_i$ on the number of context values for $t_i$. There is one atomic module for each context. We let $G$ be an integer such that $\sum_i g_i \leq G$.

**Assumption 5.5** (No distribution shift). For a latent dependency DAG and circuit for task $t_i$, suppose $t_j$ is one of the nodes in the circuit of task $t_i$, and let $\mathbf{x} \sim \mathcal{D}_i$ be the input to $t_i$. For each $\mathbf{x}_j$ computed as an input to $t_j$ when the circuit is evaluated on $\mathbf{x}$, we assume $\mathbf{x}_j$ belongs to $\mathcal{D}_j$.

Given the problem set-up above, we present our main result for this section:

**Theorem 5.6** (Learning DAG using v2). *Given a latent dependency DAG of tasks over $N$ nodes and height $\ell$, and a circuit per internal node in the DAG, there exists an architecture v2 that learns all these tasks (up to error rate $\epsilon_L$ as defined in Sec.4) with at most $O(\ell G M 2^{O(d^2 + d \log(N/d))})$ steps.*

## 5.1 CONTEXT FUNCTION AS A DECISION TREE

In architecture v2 we use a more complicated context function $f$ to extract the stable context for each task. The context function can be implemented as a modular decision tree where each node is a separate module. We are given a compound sketch $[S_1, .., S_k]$ where we assume the sketches are ordered by importance (e.g., based on frequency: if there are $m$ hash buckets we will only track contexts that appear at least $O(1/m)$ fraction of the time, while others get "timed out" – we assume $m \geq G 2^{O(d^2 + d \log(N/d))}$). We apply $f$ recursively and then over $f(S_1), .., f(S_k)$ from left to right in a decision tree where each branch either keeps or drops each item and stops or continues based on what obtains the highest rewards, tracked at each node (subtree) of the decision tree. Thus the context function can be implemented as a recursive call to a decision tree  f($[S_1, .., S_k]$) = **DecisionTree**($[f(S_1),...,f(S_k)]$) (see Alg.2 )—each node of the decision tree will be implemented in a separate module (hash bucket).

---

**Algorithm 2** Decision Tree

**DecisionTree**($[C_1, ..C_k]$) = **TreeWalk**($[]$, $[C_1, ..C_k]$)

**TreeWalk**($l, [C_i, .., C_k]$) = branch-based-on-argmax (
    reward(h($[$TREE-WALK, $l$.append($C_i$)$]$)$]$)) :**TreeWalk**($l$.append($C_i$), $[C_{i+1}, .., C_k]$) /* keep $C_i$ in $l$ and proceed to next field */
    reward(h($[$TREE-WALK, $l]$)$]$)): **TreeWalk**( $l, [C_{i+1}, .., C_k]$) /* drop $C_i$ and proceed to next field */
    reward(h($[$TREE-WALK, $l$.append(END-WALK-SYMBOL)$]$)) ) : **TreeWalk**($l, []$) /* exit the walk and output $l$ */
) /* $l$ is the subset of fields from the sketch from the prefix processed so far, $C_i, .., C_k$ is the remaining part of the sketch. $l$ is used as the context for this current decision tree node and $[C_1, .., C_k]$ is the input sketch. Each distinct value of $l$ is a separate decision tree node */

---

The branch statement is branching to a one of the three buckets: h($[$TREE-WALK, l.append($C_i$)$]$), h($[$TREE-WALK, $l]$)$]$, or h($[$TREE-WALK, $l$.append(END-WALK-SYMBOL)$]$) based on the rewards; each bucket continues the decision tree walk with the rest of the entries in the list of contexts. Note that during training the branch will be a probabilistic softmax rather than a deterministic argmax, with a temperature parameter $T$ that controls the exploration of the branches and decreases eventually to near 0; thus the probability of each branch is proportional to $e^{-R_{branch}/T}$, where $R_{branch}$ is the reward of the branch. Initially all rewards are 0 and so all branching probabilities are all equal to $1/3$ (but there could be some other priors). Over time as the temperature is lowered, the probability concentrates gradually on the bucket with maximum reward. See Appendix F for full details.

**Claim 5.7.** *If $p$ is the initial probability of taking the optimal reward path to the leaf in the DecisionTree algorithm above, there is a schedule for the temperature in Algorithm 2, so that in $O(1/p \log 1/\delta)$ tree walk steps the modules at the nodes of the tree will converge so that the decision tree achieves optimal rewards with high probability $1 - \delta$.*

*Proof.* We will keep a very high initial temperature $T$ (say $\infty$) for $O((1/p) \log(1/\delta))$ tree walk steps and then suddenly freeze it to near zero (which converts the softmax to a max) after these steps are finished. In these initial steps with high probability $1 - \delta$ the optimal path to the leaf will have been visited at least $1/\delta$ times. Since each node is tracking the optimal rewards in its subtree, the recorded best path from root will have tracked at least this optimal reward. □

## 5.2 INCREMENTALLY LEARNING A NEW NODE (IMPLICIT TASK)

In this subsection we provide an induction proof sketch for Theorem 5.6. In the previous subsection, we saw how the context function can be implemented as a probabilistic decision tree. Other functions of the routing module that involve making subset-choosing decisions (such as Lines 13 & 21

in Alg. 1): for example, selecting a subset of $d$ pre-existing modules as children of a new task in v1 can be done using a separate decision tree (e.g. Alg. 2) where one needs to select a subset of at most $d$. This again becomes very similar to the operation of the context function: we just need to input all matured modules of the previous layer to Alg. 2 and find the $\leq d$ child modules. In architecture v2 any subset-choosing decision in our architecture can be done by using Alg. 2.

The learning algorithm follows the framework of Alg.1. The circuit routing is also done by Alg. 2: we feed all the $O(\binom{N}{d}3^{\binom{d}{2}})$ candidate edges of the circuit to Alg. 2, which finds the correct subset. The inductive guarantee that lower-level tasks are well-trained comes from the bottom-up online algorithm of v1. Modules are marked as mature based on performance, and new modules are only built on top of mature previous nodes. The probability of picking the right sequence of decisions for perform the new task is $p = 1/2^{O(d^2+d\log(N/d))}$ (including which identifying which previous possibly implicit tasks it depends on and wiring them correctly with the right contexts) and it takes $M$ examples to train the task, then the task can be learned in $O(M2^{O(d^2+d\log(N/d))})$ steps per atomic module (see Appendix F for full details).

## 6 EXPERIMENTS

We empirically examine two tasks for which learning benefits from using a modular architecture in this section. We compare an "end to end" learning approach to a modular learning approach which explores a DAG of previously learned tasks probabilistically.

### 6.1 LEARNING INTERSECTIONS OF HALFSPACES

Learning intersections of halfspaces has been studied extensively, see for example (Klivans & Sherstov, 2009). We first describe the experiment setting. Let $K$ be the number of hyperplanes, $D$ feature space dimension, we generate the following data: hyperplane coefficients $w_k \in \mathbb{R}^D$, $k = 1, 2, ..., K$ whose components are independent and follow standard normal distribution; 2.feature $x_i \in \mathbb{R}^D$, $i = 1, 2, ..., N$, independently chosen uniformly from $[-1, 1]$. And we have $y_i = \prod_{k\in[K]} \text{sgn}(w_k \cdot x_i)$, where sgn is the sign function.

While learning a single halfspace $K = 1$ is easily solved by a two-layer network with ReLU activation, it becomes much more difficult for neural networks to learn when $K$ grows. This can be observed in Figure-2, where a 3-layer neural network is used to learn the intersections.

For a modular approach, we follow Algorithm-3, which is a simplified version of Algorithm–1 and it probabilistically route to sub-modules. The input data are batches of triplets $\{(k, x_i^k, y_i^k)\}_{i\in[B]}$, where $B$ is the batch size, $k \in [K + 1]$ is the task id, $x_i^1 = ... = x_i^{K+1} = x_i$, $y_i^k = \text{sgn}(w_k \cdot x_i)$ for $k \in [K]$ and $y_i^{K+1} = \prod_{k\in[K]} y_i^k$, and we maintain a task list $T$ and module list $\Phi$.

The results are plotted in Figure-2, with $K = 1, 2, ..., 10$, $D = 100$, $p_{\text{atomic}} = 0.5$ and $p_{\text{compound}} = 0.75$. For the modular approach, all the modules are 3 layer fully-connected network of the same size and are trained for 10 epochs. For the end-to-end approach, a single 3 layer fully-connected network with 10x hidden units of the modular models and is trained until convergence. We observe for large $K$ ($K \geq 7$ in the figure), the end-to-end approach fails at the task while the modular approach continues to have good performance. See appendix for more details.

### 6.2 FIVE DIGIT RECOGNITION

In this experiment, we compare the "end to end" approach and a modular approach for the task of recognizing 5-digit numbers, where the input is an image that contains 5 digits from left to right, and the expected output is the number that is formed from concatenating the 5 digits. This task is described in Example 2 of Appendix F. Note in this task, we have 3 sub-tasks: task-1 is single digit recognition, task-2 is image segmentation, and task-3 is 5 digit recognition.

For the "end to end" approach, we train a convolutional neural network to predict the 5 digit number (see appendix for more details). For the modular approach, the input data are batches of triplets $\{(k, x_i^{(k)}, y_i^{(k)})\}_{i\in[B]}$, where $B$ is the batch size, $k \in [3]$, $x_i^{(1)} = x_i^{(2)} = x_i^{(3)} = x_i$ is the image.

---

**Algorithm 3** Probabilistic routing algorithm

**Input**: Batches of $\{(k, x_i^k, y_i^k)\}_{i \in [B]}$, $k$ the same within a batch.

**Constants**: $p_{\text{atomic}}, p_{\text{compound}}$.

**Initialization**: Set of modules $\Phi = \emptyset$, set of task ids $T = [K+1]$

Repeat the following steps:

1. w.p. $p_{\text{atomic}}$, train an atomic module $\phi_k$ that maps $x_i^k$ to $y_i^k$ (note we keep a separate copy of $\phi_k$ for each different DAG structure based on iteration choices in step 2 from the original input to $x_i^k$). If training succeeds, set $\hat{\phi}_k$ to be the DAG upto $\phi_k$ and add it to $\Phi$.

2. w.p. $1 - p_{\text{atomic}}$, for each $\hat{\phi} \in \Phi$, w.p. $p_{\text{compound}}$, set $x_i^k \leftarrow$ concat$(x_i^k, \phi(x_i^k))$.

**Return** $M$.

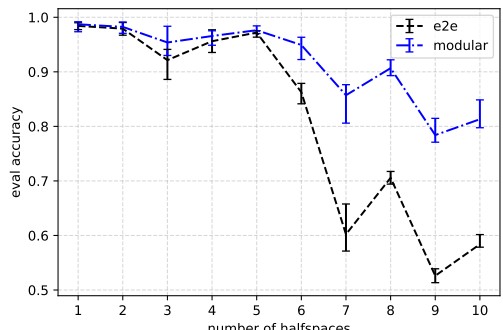

Figure 2: Intersection of halfspaces **Left:** Pseudocodes for the modular approach ("w.p." is abbreviation for "with probability"). **Right:** Modular approach continuous to have good performance while end-to-end approach fails to learn for $K \geq 7$.

| method | accuracy | steps (1-digit) | steps (segmentation) | steps (5-digits) |
|--------|----------|-----------------|----------------------|------------------|
| end-to-end | $74.5 \pm 4.5$ % | NA | NA | 18760 |
| modular | $92.0 \pm 0.5\%$ | 2560 | 640 | $18760 \pm 9380$ |

Table 1: Comparison of end-to-end and modular algorithms for 5-digits recognition: accuracy and number of training steps for different tasks to succeed. Note here each step is processing one batch with a batch size of 128, and we consider a task successful if the accuracy is above 90%.

$y_i^{(1)}$ is the single digit label, $y_i^{(2)}$ is 5 segmentation coordinate pairs (upper-left and lower-right coordinates), and $y_i^{(3)}$ is the 5 digit number label. We also maintain a task list $T$ and module list $\Phi$. For training an atomic module in Algorithm-3, we only allow the module to take inputs of the same modality (i.e. either only image or only digits, discarding the others).

We construct the training and test datasets by concatenating 5 images from the MNIST dataset. The results of the two approaches are compared in Table-1. We observe the modular approach achieves higher accuracy and has less variance with the same training steps.

## 7 DISCUSSION AND FUTURE WORK

We saw a uniform continual learning architecture that learns tasks hierarchically based on sketches, LSH, and modules. We also show how a knowledge graph is formed across hash buckets as nodes and formally show its utility (for e.g. for finding common friends in a social network) in Appendix G. Extensions of the decision tree learning to solve reinforcement learning tasks are shown in Appendix H. Although our inputs were labeled with a unique task description vector, we note our architecture works even with noisy but well-separated contexts (see Appendix D). A weakness of our work is that we have ignored how logic or language could be handled in this architecture – while perhaps there could be separate compound modules for those kinds of tasks, we leave that topic open.

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
