# OpenReview forum: "Provable hierarchical lifelong learning with a sketch-based modular architecture"
_ICLR.cc/2022/Conference — ICLR 2022 Submitted_

### Official Review · Reviewer_yxwW · 2021-11-02

**Correctness:** 4
**Technical Novelty And Significance:** 3
**Empirical Novelty And Significance:** 2
**Recommendation:** 5
**Confidence:** 3

**Main Review:**

The major selling point of this paper for me is that the authors actually provide theoretical analysis and error bounds for the learning of a modular learning strategy. While there are a number of modular approaches in the literature, I am not aware of any that have provided this type of analysis. I also appreciate that the authors try to provide analysis for more challenging settings including those where tasks depend on other tasks and no task boundaries are directly provided.

On the other hand, one negative for me about this paper was the lack of clarity, particularly when explaining what a sketch actually is. The authors try to use very general wording in some places, but for me it began to really loose any meaning. I think it would be nice to at least walk through exactly how this all works in one example. Another thing that could really improve the paper in this regard would be to provide concrete connections to past modular architectures in the literature. In particular the following papers seem relevant to contrast with:

"Routing Networks: Adaptive Selection of Non-linear Functions for Multi-Task Learning" Rosenbaum et al., ICLR-18
"Modular Networks: Learning to Decompose Neural Computation" Kirsch et al., NeurIPS-18
"Automatically Composing Representation Transformations as a Means for Generalization" Chang et al., ICLR-19
"Recursive Routing Networks: Learning to Compose Modules for Language Understanding" Cases et al., NAACL-19
"Routing Networks and the Challenges of Modular and Compositional Computation" Rosenbaum et al., 2019

In comparison to the above papers, for example, the domains considered in this paper are quite toy and minimal. While I appreciate that the authors also provide theoretical results, it is hard to get much out of these experiments. Additionally, even beyond the environment complexity considered and variety of domains, there are not many interesting models considered in each experiment either. There are a bunch of potential modular learning baselines that could be considered. Moreover, even just considering ablations of the proposed approach i.e. the choice of hashing etc would be an improvement.


**Summary Of The Paper:**

The authors propose a modular architecture for multitask learning in the presence of hierarchically structured tasks leveraging sketches and locally sensitive hashing. They further provide proofs and error bounds about how the proposed architecture can learn tasks while leveraging access to previously learned subroutines. The authors additionally provide extensions to the proposed architecture in the case of tasks depending on other tasks and lack of clear task boundaries. Empirical studies are conducted for this modular approach in comparison to end-to-end models on synthetic domains related to learning the intersection of half-spaces and MNIST digits.

**Summary Of The Review:**

I think there are some major positive aspects to this paper including that the authors provide theoretical analysis of their modular approach, which is rare in the literature. Additionally, the authors discuss extensions of the approach to more ambitious settings. On the other hand, there is lack of clarity in the writing in some places and the paper lacks positioning with respect to existing modular approaches in the literature. The empirical evaluation of the approach is very limited in comparison to prior modular architectures, which makes me question if there are some undisclosed scalability concerns. As such, I lean towards rejection at this time.

---

> ### Author Response · Authors · 2021-11-16
> **Response to Reviewer yxwW**
>
> Thanks for pointing out the related modular works -- they are all very nice papers but broadly, the difference is that none of them provide any theoretical guarantees. The most closely related is “Modular Networks: Learning to Decompose Neural Computation” which is conceptually similar to ours, but again our novelty is that we provided theoretical analysis and identified difficulties and sufficient conditions for such tasks. In the experiments, we are just showing that the End-to-End model fails even in these toy problems whereas our approach succeeds. This demonstrates that even in simple, non-pathological cases the modular approach may be substantially more effective.

---

### Official Review · Reviewer_RKp7 · 2021-11-02

**Correctness:** 3
**Technical Novelty And Significance:** 2
**Empirical Novelty And Significance:** 2
**Recommendation:** 3
**Confidence:** 3

**Main Review:**

############## Strengths ##############

1. The problem at hand, modular lifelong learning, is of interest to the lifelong/continual learning community
2. This is one of the few works that addresses lifelong learning from a theoretical perspective

############## Weaknesses ##############

1. The paper is organized in a way that is very hard to follow, and in particular the majority of the technical details are left to an 18-page appendix
2. The theoretical results appear to be minor
3. The empirical analysis is minimal


############## Arguments ##############

I believe that the line of work proposed in this paper is quite interesting and relevant: modular lifelong learning from a theoretical perspective.

However, I had a very difficult time following the submission. The authors start with a subsection of the introduction providing a large set of detailed technical descriptions of the approach before even introducing the concepts or the motivation behind them. I think it would be best if this section instead started with just high-level explanations based on "functions", "pointers", and "arguments", with only a mention about how sketches and LSH are the mechanisms to implement these ideas. Then, after the reader is comfortable with the overall ideas, the formal concepts may be introduced and then the technical description can be made in terms of these formal concepts. In particular, a one sentence summary of (my understanding of) the approach is that each input is hashed and this hash is used to select the module that will be used to process the input. This is not fundamentally different than e.g. gating nets or any sort of clustering-based routing. The whole paper could be made much clearer by providing intuitive descriptions ahead of the technical details. More critically, the manuscript repeatedly uses technical descriptions of e.g. sketches as a means for providing intuition about the approach, but I believe that sketches should be used as only a tool for deriving results and the concrete architecture, but not as a means for providing intuitive descriptions: modularity can be understood very well without the introduction of sketches. Moreover, this repeated use of sketches takes up a lot of space for explaining insights and intuitions that are _not_ part of the contributions of this work, but of Ghazi et al. (2019). Instead, a concrete, brief definition of sketches should be given, and then the intuitions be built more broadly. Incidentally, for all the details, properties, and examples given, sketches are never actually defined. I thought they were their dense representations, but the end of Section 2 indicates that they are _converted_ into dense representations. It seems like therefore they are actually data structures like sets, trees, tuples... Or maybe I'm wrong and sketches are indeed vectors and the compound structures are built as described in the recursive computation.

In terms of the actual contributions, I am concerned that the theoretical results obtained are very minor. At a high level, in my understanding, the authors make a large set of assumptions that in summary enable the agent to learn near-perfect models for each task and perfect combinations of modules via exhaustive search. It is unclear how much insight this provides, and the technical tools used for showing these results (sketches) are not a contribution of this paper. More concretely:
- From the problem definition, it seems that if there are N possible tasks, then at each time step one of these N tasks is uniformly sampled. This means that over the O(MN log N/delta) steps, each tasks is seen ~M log N/delta times. This doesn't really seem to match any common lifelong learning formulation, but instead seems closer to an interleaved version of multi-task learning. While Ruvolo and Eaton (2013) also assumed that tasks were drawn in some stationary fashion from a distribution over tasks, they did _not_ assume a fixed set of N tasks, and therefore the probability of task repetitions was negligible (and, in practice, in their experiments tasks were never repeated).
- In case of the DAGs, the authors show that if the tasks indeed form a DAG and the agent is allowed to keep track of all possible DAGs with hashing and context functions that perfectly capture task relations, then the agent is able to provably learn to solve the tasks. Plus, there is not even a true lifelong setting. These assumptions seem to oversimplify the problem to the point that any insights gained from these results are not really useful for any realistic setting.

This wouldn't necessarily be a problem if the submission included substantial empirical evaluations that validated the proposed approach. However, the experiments fall short from a proper empirical evaluation for a lifelong/continual learning paper. The authors consider only two toy problems and show that their method is capable of working better than a vanilla non-modular approach. Given the fact that the theoretical results appear to be relatively minor, I would encourage the authors to carry out a more complete empirical evaluation, similar to that of other lifelong learning works.

############## Additional feedback ##############

The following points are provided as feedback to hopefully help better shape the submitted manuscript, but did not impact my recommendation in a major way.

Intro
- So far the proposed problem is intriguing
- Missing cites to PAC lifelong learning [1] and other compositional/modular lifelong learning works [2, 3]

Sec 1.1
- The discussion of memory over programs vs data being artificial seems to be very tangential to the point
- I'm not sure I get the purpose of the comparison to scene segmentation. Many of the cites and remarks seem pretty arbitrary
- I have a lot of faith in the contributions this paper will produce so far, but it almost feels like it was written deliberately to confuse readers. It's not poor language or grammar at all, just completely unclear structure and a level of technicality that seems to bury intuitions and motivations so deep that they're almost impossible to uncover. Should have started with the "overall loop" intuition, then inputs/outputs, then hashes... but it's all flipped around, there's a lot of back and forth, ideas repeated in different orders at different points...

Sec 2.1
- No intuition is given here about how sketches to combine into a compound sketch might be chosen. This is likely chosen somehow based on "frequent subsets" as suggested in the penultimate paragraph of page 2, but this is so far not explained or motivated.

Sec 3
- Last paragraph on page 4: shouldn't it be "projects the sketch down to the task descriptor [s_t]"?
- Before stating the algorithm and claim, the authors should describe in plain language what the meaning of choosing f(x_t,y_t,s_t) = s_t is. I believe it simply says that there will be one independent function per task (as suggested by the Section heading), but this is never stated explicitly. This is particularly unclear because the hash function and the task descriptor are not defined, so it's unclear whether "similar" tasks would also be mapped to the same function, nor what "similar" tasks would even mean in this context.

Sec 4
- Authors state that "like in section 3" they use any LSH, so one might assume that this is also the case in Sec 3, although it wasn't stated

Appendices
- The appendices are incredibly long (18 pages!) and the authors claim that they contain most of the details omitted in the main paper (like what the actual architectures are and so on). I did not read the appendices in detail.
- Beside these, they include seemingly tangential sections like the knowledge graph, connections to RL, a card-game case study, and even a "Misc" appendix. Given the overall length of the paper+appendices, it would perhaps be useful to consider cutting these tangential discussions.

Typos and grammar
- Sec 2, par 3: a sketch chat can... --> a sketch that can...
- Sec 2, par 3: "However the sketch may be more complicated like an object for example the <person-sketch>could in turn be set of such pairs f[NAME,<name-sketch>], [FACIAL-FEATURES,<facial-sketch>], [POSTURE,<posturesketch>]g." --> this is a run-on sentence and has missing words and spaces
- Throughout Sec 2 there are missing words and periods, and grammar errors like subject-verb agreement.


[1] Pentina and Lampert. A PAC-Bayesian bound for lifelong learning. ICML 2014

[2] Mendez and Eaton. Lifelong learning of compositional structures. ICLR 2021

[3] Veniat et al. Efficient Continual Learning with Modular Networks and Task-Driven Priors. ICLR 2021


**Summary Of The Paper:**

The submission proposes an approach to lifelong learning using a sketch-based modular architecture. The primary contributions of the paper are proofs of the learnability of certain classes of tasks with three different choices of modular architectures. The algorithms are also evaluated empirically on toy supervised learning tasks.


**Summary Of The Review:**

Unfortunately, I recommend the rejection of this work. The submission lies at the intersection of theoretical and empirical research, yet falls short in both these axes in terms of results. In particular, the theoretical results seem sufficient for an approach that is empirically applicable to realistic problems, yet is evaluated on a set of simple toy problems only. On the other hand, the empirical results on toy data seem sufficient for an approach for which there are major theoretical results, but the theorems in the submission are, in my understanding, fairly minor. The manuscript's structure is quite odd, which makes it hard to follow and contextualize the contributions, so substantial editing might change my overall perception of the work.

---

> ### Author Response · Authors · 2021-11-13
> **Response to Reviewer RKp7**
>
> We appreciate the reviewer's comments.
>
> 1. The Ghazi et al paper gives a way of sketching the execution of a deep network assuming it is modular but does not show how the modular network is built in the first place. This work shows how to build modules hierarchically in parallel with storing/retrieving sketches from an LSH table (see sketch memory paper too for empirical validation). Indeed trees and dags are structured data that are ``converted’’ into dense representations so as to be able to apply standard LSH functions. Further, the structured to dense conversion is reversible (provided the data is small to fit all the information in the dense sketch representation). Thus the dense format is useful and equivalent to the structured format of a sketch.
>
> 2. The hierarchical learning problem is interesting when the data do not come in topologically sorted order. So assuming i.i.d. task arrival is just a stylized assumption that captures this situation. As a matter of fact, a regular lifelong learning setting such as Ruvolo and Eaton (2013) assumes the data for each task to come in batches, this means there is already repetition of task data within the batch. Such a batch arrival assumption is no more realistic than assuming tasks arrive at random (please also see our response bullet point 1 to Reviewer kTWq).
>
> 3. Hashing and the context function do not perfectly capture the relationships between tasks. In v0 and v1, they provide nothing more but task IDs indicating which data should be grouped together into a task by the algorithm. We need to discover the hierarchical structure by ourselves. We also argue even the assumption in v0 and v1 that the task IDs are given is still too strong, e.g., for an autonomous lifelong learner. Indeed, v2 starts departing from this assumption -- the continuously learned modular structure is itself used to figure out the real “context/task” that is used to map data to a bucket or a concept. This has not been studied in previous lifelong learning works, in which task IDs are given along with the batch of task data. We also concretely point out how new concepts may be automatically discovered by finding a new LSH bucket.  We don’t keep track of all possible DAGs in the algorithm. The “DAG” of task dependency hierarchy is different from the “circuit” in v2. The circuits help identify the task ID. We track all the circuits (ways of composing the children) for each task, whose size is controlled by the small number of its submodules/children. To incrementally learn the DAG, for each task we only keep track of the set of submodules (children), whose size is again sufficiently small.
>
> 4. We are just showing that the End-to-End model fails even in these toy problems whereas our approach succeeds. This demonstrates that even in simple, non-pathological cases the modular approach may be substantially more effective.

---

> > ### Comment · Reviewer_RKp7 · 2021-11-17
> > **Response to authors' comments**
> >
> > I appreciate the authors responding to my concerns. Unfortunately, my concerns do not seem to have been addressed, and these concerns seem to match other reviewers' concerns.
> >
> > 1. Thank you for giving some description of how this submission differs from the Ghazi et al. paper. However, these do not seem to be enough details to differentiate this contribution from theirs. What is the difference between "sketching the execution of a deep network" and "build the modular network"? Similarly, the authors somewhat confirm my confusion between the "structural" and "dense" sketch definitions. But again, what _concretely_ is the definition of a sketch? This definition is still lacking from the paper and even from the authors' response. Given that all reviewers commented on the lack of clarity throughout the manuscript, I would strongly encourage the authors to apply major revisions to their draft. If they are able to do so within the next few days prior to the end of the discussion period, I would be happy to look over the revised manuscript. However, I cannot increase my score for the paper in its current form. Note that these two points that the authors addressed are _not_ the only major concerns about clarity.
> > 2. What do the authors mean with "topologically sorted order"? Given their responses to other reviewers, I can guess that they mean the "hierarchical dependency order". If this is the case, I am not sure why this point is relevant: most orders are _not_ the hierarchical dependency order. I might agree with the authors that potentially the setting without task revisiting is not necessarily more realistic than other settings, but given the conventions that have arisen in the past few years around the "lifelong learning" terminology, it seems deceiving to denote the setting in this submission as lifelong learning.
> > 3. The distinction between "task descriptor" and "task ID" is quite relevant. Section 3 uses "task descriptor" both to describe s^t and t, which causes confusion. Still, when the task ID is given architecture v0 is trivial, and v1 becomes trivial if the agent can keep track of many (d-1) choices of dependencies (as I believe this approach does). Given the descriptions in this response and in the manuscript, it is still unclear what the difference is between the DAG and the circuit: the circuit is described as containing all the other tasks that the current task depends on. This needs to be explained a lot more clearly.
> > 4. I understand that the purpose of this work is to provide theoretical results for modular architectures. What I meant to convey (and I believe this was stated clearly enough in the initial review) is that given the (apparently) relatively low value of the theoretical contributions, this work would benefit from substantially more extensive experiments. Alternatively, the authors could more clearly delineate the theoretical contributions to highlight their value.

---

> > > ### Author Response · Authors · 2021-11-23
> > > **Follow-up response to Reviewer RKp7**
> > >
> > >  1. Thanks for pointing out the confusion. “Sketching the execution of a deep network” means concisely encoding the computation network in a dense vector from which the entire computation can later be retrieved, thus permitting interpretable embedding. However, in the previous work of Ghazi et al. 2019, this process requires the computation network to be given beforehand. In our case, “build the modular network” means we construct these sketches without knowing the network, but discover it by ourselves. We rewrote the sketch review section to include a clearer description of what a sketch is.
> > >
> > > 2. We agree that in conventional lifelong learning where the purpose is to discover shared latent features, the data is not given in hierarchical dependency order (if that is what you meant). But this is because conventional lifelong learning does not concern potential hierarchical interdependency between tasks.  While we acknowledge that this setting is very unconventional, we believe that real-life learning does involve building skills/ concepts in a hierarchy over one’s lifetime. For example, as we’ve shown in our experiment, the task of recognizing 1 digit numbers significantly improves the task for 5 digit numbers, this is a very natural setting of the human learning process where we would counter both tasks randomly early in our life.
> > >
> > > 3. Indeed, Sec. 3-4 might seem trivial, but that’s only when we know which task each data sample belongs to. We illustrate the simple problems first to build up to the harder problem in Sec. 5., where the circuit identifies the task (given a vague task descriptor).  The dependency DAG simply tells you for each task, which d subtasks we should use.  However, it’s possible that different partial orderings of performing these subtasks -- ie., the way in which the functions are composed -- can define very different task functions. Therefore, we need to learn a separate circuit to capture each different partial orderings.
> > >
> > >
> > > 4. The theoretical contribution is as follows. We are studying an unconventional lifelong learning problem with a modular hierarchy, where the data is arriving in a more realistic way. Within this setting, the use of decision trees as a control structure for choosing among candidate structures for the composition of modules is one contribution we wish to stress: note that the choices of alternative structures is combinatorial and not differentiable. The theoretical guarantees that form a major part of the total contribution are also novel: it is not immediately obvious what conditions are sufficient to enable hierarchical composition, on account of the accumulation of error and so forth.

---

### Official Review · Reviewer_kTWq · 2021-11-02

**Correctness:** 2
**Technical Novelty And Significance:** 2
**Empirical Novelty And Significance:** 2
**Recommendation:** 3
**Confidence:** 2

**Main Review:**

## Strengths

1. The paper looks at "lifelong learning" from a theoretical perspective. I think the community will benefit from more works in this research direction.

2. While the paper focuses on theoretical contributions, it does include some empirical results to show that the proposed approach works in practice.

## Areas for improvement

1. The problem is not well scooped. The paper is focusing on the sequential multi-task learning where the agent can come across a task multiple times. One can argue that this is also a form of lifelong learning (and while I do not have a strong opinion here, I am okay if the paper wants to consider this setup as a lifelong learning setup). Contrast this with the more common lifelong learning setup where the agent does not see the same task multiple times and has to deal with challenges like catastrophic forgetting. Note that I am not arguing that one problem (instance) is better/more important than the other. I am highlighting that it is important to highlight the setup (instead of pitching the work as the general lifelong learning setting).

2. Very limited related work - I understand that the paper is focusing on hierarchical task distributions but there is much more relevant work than just the 2 papers cited on page . eg https://sites.google.com/view/cl-theory-icml2021 Similarly, RL has a rich literature on hierarchical learning (eg Options framework, Feudal Networks etc). Work on modular networks should be cited eg https://arxiv.org/abs/2012.12631. Other works on lifelong learning should also be cited eg: https://www.sciencedirect.com/science/article/pii/S1364661320302199 (these are pretty much the first works that came up on a quick google search).

3. The paper is hard to understand with many important details/definitions missing. It will help with Section 2 is moved before section 1.1 There are several assumptions/statements but their role is not clear. For example, it is assumed that the tasks are sampled uniformly and datapoints are sampled using a fixed distribution from the task. What is the nature of this "fixed distribution" Does the nature matter or it can be any fixed distribution? What parts of the algorithm depend on this assumption. What happens if the task distribution is not stationary (but data distribution within each task is stationary) or the other way round. Since the paper is focusing on the theoretical aspects, it is important to explain/outline the effect of these asumptions.

4. I am not sure of the novelty and significance of the paper. A lot of previous works have looked at the idea of using modularity for in context of hierarchal/multi-task/compositional learning https://arxiv.org/pdf/1611.01796.pdf, learning composable modules/representations https://arxiv.org/abs/1807.04640 or routing information/representation through modules / using modules recursively: https://openreview.net/forum?id=mLcmdlEUxy-

5. It is not clear how to use the proposed approach in practice. I understand that the work is focusing on theory (and I am not asking for non-toyyish experiments) but it is not clear as to how obtain the atomic tasks (to train the atomic modules) given standard tasks. Moreover, for half-spaces and five digit recognition, it is not clear how well do standard modular networks perform, just limiting the value of empirical results.

**Summary Of The Paper:**

The paper proposes a modular architecture for lifelong learning of hierarchically structured tasks with some theoretically guarantees. The paper includes some experiments, on simple domains, with a single baseline.


**Summary Of The Review:**

Please note that my initial review of the paper is based on what I understood so far. I look forward to the author's response and interacting with them to understand their approach better. I encourage them to initiate a conversation sooner rather than later. I would be very happy to change my scores as I better understand the contribution/significance of the work.

In the current stage, the paper is missing several important details, making it hard to understand if the paper is novel/significant or not. Several important references are missing and it is not clear how does the method compare against standard modular networks.

---

> ### Author Response · Authors · 2021-11-13
> **Response to Reviewer kTWq**
>
> We appreciate the reviewer's comments.
>
> 1. Indeed, the problem we are focusing on is different from the formulation used in previous lifelong learning works. You may think of our setting as the data for different batches arriving online and in random order. We believe that in practice, it’s not more realistic to get one batch of data for each task and never see the same task again. If possible, we would like to ask why is such batch data arrival setting more natural? In any case, one of the main challenges we seek to address in our work is to identify the dependency structure among the tasks. This problem would be much less interesting if the data were presented in batches in the dependency order.
> Note that we are not overwriting the model, but we allocate the model through hashing and freezing modules, which gives us a method that avoids catastrophic forgetting which could be a potentially useful architectural component in practice.
>
> 2. Thanks for pointing these out. We will certainly include these citations.
>
> 3. We appreciate the suggestion of rearranging the presentation. But first, we would like to ask which parts seem to have missing definitions, so that we can provide them. Regarding the fixed distribution: for generalization, the distribution cannot be totally arbitrary. Later tasks’ data distributions have to somehow relate to previous tasks’ data distributions. For the sake of keeping the theoretical proof simple and not complicating the main point, we make the fixed distribution assumption. It is easy to extend this to the case for example when the distribution’s density function changes by some constant factor over time.
>
> 4. The novelty is that we have provided theoretical proofs, and identified sufficient conditions for learning these hierarchical tasks in an online setting. Our theoretical analysis highlights the role of several key architectural features, specifically the LSH table, context function, sketches, decision tree combining combinatorial search with deep-learned modules -- this careful combination of architectural features with theoretical guarantees is also part of the novelty.
>
> 5. First, we would like to ask: what do you mean by “standard tasks”? Regarding the training method of the atomic module: we use robust SGD, such as (Diakonikolas et al., 2019), which we mentioned at the end of the first paragraph in Section 4.

---

> > ### Comment · Reviewer_kTWq · 2021-12-01
> > **Thank you for your response**
> >
> > I thank the authors for their response. However, most of the questions I (and other reviewers) raised do not seem to be answered. I will consider changing my review based on the discussion with the other reviewers.
> >
> > 1. I suggest highlighting the specific setup that the paper is focusing on (instead of pitching it as a general lifelong learning setup) and I clearly state that "I am not arguing that one problem (instance) is better/more important than the other. ". The authors can still argue that their setup is more general but it is one of the lifelong learning setups.
> >
> > 3. `Regarding the fixed distribution: for generalization, the distribution cannot be totally arbitrary`. Makes sense - so what is this distribution? Other questions like "Does the nature matter or it can be any fixed distribution? What parts of the algorithm depend on this assumption. What happens if the task distribution is not stationary (but data distribution within each task is stationary) or the other way round. Since the paper is focusing on the theoretical aspects, it is important to explain/outline the effect of these assumptions." are not answered.  `we would like to ask which parts seem to have missing definitions` - Section 1.1 spends good amount of time on "sketch" while it is not clear what a sketch is. The defn comes in section 2.
> >
> > 4. Will go through this part of the paper again to understand the novelty.
> >
> > 5. A standard task is any task - e.g. the task of making coffee in the morning. How does the system obtain atomic tasks for this task?

---

> > > ### Author Response · Authors · 2021-12-01
> > > **Followup response to kTWq:**
> > >
> > > We appreciate the comments. In the following, we give further explanations in the hope of clarifying the confusion.
> > >
> > > Regarding bullet point 2:
> > >
> > > 1. For simplicity we assumed that the task distribution is iid (stationary). We actually only need that because the inputs for a higher-level task cannot all come before the inputs of a lower-level task. As long as enough training data for the higher-level tasks come after that of the lower tasks, our architecture works to learn these tasks. Our iid assumption on tasks was only a simplification for ease of exposition.
> > >
> > > 2. Stationary distribution within each task is assumed to avoid problems of distribution shift and to easily apply generalization bounds. Take the 5-digit modular learning, for example, one of its lower-level tasks is learning to identify 1 digit numbers. However, the input to 1-digit recognition task is different from the input to the 5-digit recognition task. Fixed distribution avoids such distribution shift. However, it’s easy to relax these assumptions slightly with weaker guarantees.
> > >
> > >
> > > Regarding bullet point 4: This is an excellent question! In fact, while writing the paper we went over several such questions including "How do we wear a shirt", "How to unscrew and open the lid of a jar". We ended up putting the "making next move in a card game" in the appendix section I.
> > >
> > > 1. Instead of looking at the task of "making coffee", let's do "making tea" instead. In order to "make tea", one has to first recursively learn tasks such as "boil water in a pan", "add tea leaves to water" "add milk to pan". Thus the task "make tea" is hierarchically built on top of these previously learned tasks (they need not be simple atomic modules) executed in that order. We believe that in general any new skill/task that we learn is built upon previously learned simpler tasks; that is, it is a "composition" of previously learned operations.
> > >
> > > 2. Data for tasks such as “boil water in a pan”, “add tea leaves to water” are labeled and come with task descriptions in the input stream.
> > >
> > > 3. Note that the task description  "boil water in a pan" doesn't point to a unique atomic module for that phrase but the actual execution pathway depends on how that description is handled through the entire previously learned set of modules to produce the right set of "sketches" that describe the action for actually boiling water in a pan.
> > >
> > > 4. In v2 we show how the atomic modules are not in one-to-one correspondence with the external task descriptions. In this way, over time the task with the description "make tea" gets built as an "execution pathway (mini-circuit)" over these simpler task descriptions. Version v2 provides a simplified caricature/abstraction of how this happens.

---

### Official Review · Reviewer_GsU1 · 2021-11-05

**Correctness:** 3
**Technical Novelty And Significance:** 3
**Empirical Novelty And Significance:** 2
**Recommendation:** 5
**Confidence:** 2

**Main Review:**

**Disclaimer**

Although I have built quite an expertise in continual and deep learning, my understanding of learning theory and theoretical computer science is still pretty shallow. For this reason, I did not fully understand the ideas presented in the paper. I will thus provide a pretty superficial review along with a low confidence score.

**Strengths**

The authors provide some algorithmic theoretical guarantees in a field where they are mostly absent, namely continual learning

**Weaknesses**

 I have found the architecture quite involved to understand. If the paper is addressed to researchers working on learning theory and/or sketches, then this is probably fine.

If however, the intended audience is your average continual-learning researcher, I think that the manuscript need works. Although most of the introduction was easy to follow, I found the remaining sections cumbersome (again, it could be because of my background). I found that section 3,4,5 bombards the reader with a plethora of architectural components, without ever providing some kind of intuitions on why any of them are important for a continual-learning system. My perspective is thus: if the main subject is continual learning, then the manuscript needs a major revision.

Lastly, the continual-learning aspect of the experiments escapes me. If they are continual-learning experiments, the authors should clearly explain the how the data distribution is non-stationary.


My perspective is that

**Summary Of The Paper:**

This work introduces a continual-learning architecture that combines elements for sketches, hash functions, modules and routing.
Theoretical guarantees are provided for the performance of the architecture.
Some experiments seem to support the advantage of the proposed architecture over conventional deep learning ones.

**Summary Of The Review:**

See Disclaimer above

---

> ### Author Response · Authors · 2021-11-16
> **Response to Reviewer GsU1**
>
> We thank the reviewers for the comments.
>
> 1. Intuitions for architectural choices in sec 3,4,5: Sec 4: we introduce these architecture components because we are trying to discover potential hierarchical relationships among the learning tasks -- here, using the learned task functions as additional features in other tasks -- in contrast to conventional lifelong learning methods where the goal is to learn common features. One way of viewing the difference is that we obtain deeper representations as a consequence of the composition of these functions, whereas typical methods learn features of a fixed, limited depth. In our examples, we show that using this hierarchical structure is important. Sec.3 just illustrates how the architecture partitions the data into tasks, for which we individually train building blocks (atomic modules). This basic model is elaborated upon in Sec.4.  Sec.5 considers the lifelong learning problem faced by an autonomous agent, in which task IDs are not provided: the agent must identify the decomposition into tasks on its own. So the v2 architecture introduces a circuit for each task that can identify its task ID.
>
> 2. In the experiments we are simply showing that the End-to-End model fails even in these toy problems whereas our modular approach succeeds, thus motivating our architecture. In our entire paper (including theory and experiments) the data is interleaved across tasks: we are considering a different setting than conventional batch-data lifelong learning. So, in general, each task is associated with its own data distribution, and the non-stationarity essentially corresponds to the different task data distributions (of course, each task has its own associated function to learn as well). We aim to perform well on all of these task distributions. Note that the i.i.d. assumption of task arrival is just a stylized model; our results do not rely on it and we could have considered a more general setting, but we are just using it to simplify the exposition. We stress that for the problems we wish to study, in which the learner must identify the hierarchical dependencies among the tasks, having the batches interleaved in this way is more challenging than if the tasks arrive in an order that respects the hierarchical structure (thus allowing a conventional lifelong learner to succeed), since then there is no question of which task should be trained first, which ones may depend on which others, and so on.

---

> > ### Comment · Reviewer_GsU1 · 2021-11-26
> > **response**
> >
> > thanks for the clarifications.
> >
> > You should add some of these explanations in the paper.
> > Again, if the intended audience is continual-learning practitioners, I think a lot of contrasting with classical CL needs to be added.

---

### Decision · Program_Chairs · 2022-01-20

**Decision:**

Reject

**Comment:**

This paper develops an approach to modular lifelong learning over hierarchical tasks, proving the learnability of certain task classes under different modular architectures, with empirical evaluations on toy supervised tasks. The authors are to be commended for being one of the few works that develop lifelong learning theory. However, the reviewers found the theoretical contributions to be relatively minimal and that the empirical work needs to provide more substantial insight before it is ready for publication. Moreover, the reviewers had substantial concerns with the paper's overall presentation, in many cases finding the paper's organization confusing with many asides and critical details relegated to the appendices. The confusing presentation especially needs to be remedied, and the authors are advised to take the reviewers' concerns into consideration when preparing future versions of their manuscript.

On a minor point, the reviewers identified several places where the paper didn't cite or develop connections to relevant current literature. The authors might also be interested in connections to some much earlier work by Utgoff and Stracuzzi on many-layered learning (Neural Computation 14.10, 2002), which shares some high-level similarities to ideas explored in this paper.